# Modifiable and Non-Modifiable Risk Factors for Atherothrombotic Ischemic Stroke among Subjects in the Malmö Diet and Cancer Study

**DOI:** 10.3390/nu13061952

**Published:** 2021-06-06

**Authors:** Anna Johansson, Isabel Drake, Gunnar Engström, Stefan Acosta

**Affiliations:** 1Department of Clinical Sciences in Malmö, Lund University, Jan Waldenströms gata 35, SE 205 02 Malmö, Sweden; isabel.drake@med.lu.se (I.D.); gunnar.engstrom@med.lu.se (G.E.); stefan.acosta@med.lu.se (S.A.); 2Vascular Center, Department of Cardiothoracic and Vascular Surgery, Skåne University Hospital, Ruth Lundskogs gata 10, SE-205 02 Malmö, Sweden

**Keywords:** ischemic stroke, atherothrombotic, epidemiology, risk factors, lifestyle, diet, physical activity

## Abstract

Risk factors for ischemic stroke is suggested to differ by etiologic subtypes. The purpose of this study was to examine the associations between modifiable and non-modifiable risk factors and atherothrombotic stroke (i.e., excluding cardioembolic stroke), and to examine if the potential benefit of modifiable lifestyle factors differs among subjects with and without predisposing comorbidities. After a median follow-up of 21.2 years, 2339 individuals were diagnosed with atherothrombotic stroke out of 26,547 study participants from the Malmö Diet and Cancer study. Using multivariable Cox regression, we examined non-modifiable (demographics and family history of stroke), semi-modifiable comorbidities (hypertension, dyslipidemia, diabetes mellitus and atherosclerotic disease), and modifiable (smoking, body mass index, diet quality, physical activity, and alcohol intake) risk factors in relation to atherothrombotic stroke. Higher age, male gender, family history of stroke, and low educational level increased the risk of atherothrombotic stroke as did predisposing comorbidities. Non-smoking (hazard ratio (HR) = 0.62, 95% confidence interval (CI) 0.56–0.68), high diet quality (HR = 0.83, 95% CI 0.72–0.97) and high leisure-time physical activity (HR = 0.89, 95% CI 0.80–0.98) decreased the risk of atherothrombotic ischemic stroke independent of established risk factors, with non-significant associations with body mass index and alcohol intake. The effect of the lifestyle factors was independent of predisposing comorbidities at baseline. The adverse effects of several cardiovascular risk factors were confirmed in this study of atherothrombotic stroke. Smoking cessation, improving diet quality and increasing physical activity level is likely to lower risk of atherothrombotic stroke in the general population as well as in patient groups at high risk.

## 1. Introduction

During the 21st century cardiovascular disease (CVD) prevalence rates have fallen in most European countries, reduced smoking and improved medical risk factor control being the main reasons [1]. However, there has been a significant increase in obesity and diabetes mellitus rates worldwide in the last decade [2], a trend that threatens to counteract the CVD decline [3]. Diabetes causes various vascular changes that can lead to major clinical complications, such as stroke [4]. Ischemic stroke is traditionally divided into large vessel disease, small vessel disease, cardioembolic, other and unknown etiology [5]. The majority of ischemic stroke cases are caused by complications to atherosclerosis such as local cerebral atherothrombosis or embolization from a thrombosis within a carotid artery plaque [6]. Established risk factors for ischemic stroke include age, male sex, hypertension, diabetes mellitus and high cholesterol. Diet, physical activity, smoking and alcohol consumption make up some of the most important modifiable risk factors [1]. While there is a large body of studies examining the impact of risk factors for all ischemic stroke, including those caused by cardiac embolization due to atrial fibrillation, fewer studies have examined how these risk factors associate with risk for incident atherothrombotic stroke. A prospective cohort study on 14,488 individuals found that the effect of different risk factors on the incidence of ischemic stroke varied across subtypes. Diabetes and current smoking were associated with both lacunar and non-lacunar ischemic stroke subtypes, with a slightly higher risk for lacunar compared to non-lacunar stroke. Higher waist to hip ratios and lipoprotein (a) levels were associated with increased risk for non-lacunar stroke, and higher blood levels of von Willebrand factor increased the risk for cardioembolic stroke [7]. These results indicate differences in risk profiles according to subtype of ischemic stroke.

Several observational studies have reported that an overall healthy lifestyle reduces the risk of ischemic stroke using lifestyle scores including non-smoking, absence of overweight, moderate physical activity and alcohol consumption, and a high diet quality [8,9]. A previous study among Swedish men found that a healthy lifestyle similarly reduced risk among men with a history of hypertension, high cholesterol levels, diabetes, heart failure or atrial fibrillation [10]. Similarly, a recent study on the UK Biobank suggested a strong benefit of a healthy lifestyle in reducing risk of stroke independent of underlying polygenic risk [11]. However, few studies have investigated the impact of specific lifestyle factors among patients with and without prior comorbidities such as hypertension, dyslipidemia, diabetes mellitus, atherosclerosis and the risk of atherothrombotic stroke. For primary prevention in a clinical setting, it is important to understand the risk benefit of adherence to favorable lifestyle habits among at-risk patient groups.

The purpose of the present prospective population-based cohort study was therefore to study the association between modifiable, semi-modifiable, and non-modifiable risk factors and incident atherothrombotic ischemic stroke unrelated to atrial fibrillation or flutter. A secondary aim was to examine if the association between modifiable lifestyle risk factors and atherothrombotic stroke differ among subjects with and without prior comorbidities, and to what extent modifiable risk factors contribute to improved prediction in addition to established risk factors.

## 2. Materials and Methods

### 2.1. Study Population and Data Collection

Men and women in the ages 46–73 born in Malmö were eligible to enter the Malmö Diet and Cancer Study (MDCS) cohort, taking part in baseline examinations between 1991–1996 [12]. The cohort includes 30,446 participants followed until the end of 2016. Participants with prevalent atrial flutter or fibrillation (AF), ischemic stroke (any type) and subjects with missing data on included covariates were excluded, after which 26,549 remained to be included in the present study (Figure 1). The study was conducted ethically in accordance with the World Medical Association Declaration of Helsinki and the protocol was approved by the Regional Ethical Review Board in Lund, Sweden (Dnr § LU 51-90, 2007/166). All subjects gave their written and oral informed consent for inclusion before they participated in the study.

### 2.2. Endpoint Ascertainment

The Swedish National Patient register and the Cause of Death Register Participants were used to identify participants with a first registered diagnosis of ischemic stroke via civic registration numbers. Diagnoses were coded using the Swedish revision of the International Classification of Disease (ICD), version 8 (433, 434), version 9 (434, 436), and version 10 (I63, I64). Patients registered with AF prior to or simultaneously (±30 days) to ischemic stroke were labeled as AF-related ischemic stroke and were not included as atherothrombotic stroke. Patients with AF-related ischemic stroke were followed up until date of incident AF. AF was ascertained by ICD8-427.9, ICD9-427D and ICD10-I48 codes. There were 312 prevalent and 4,511 incident AF during follow up, and the number of incident ischemic strokes was reduced from 2,847 individuals unadjusted for AF to 2339 individuals adjusted for AF.

### 2.3. Validation of Ischemic Stroke Diagnosis

Validity of diagnosis of ischemic stroke in the national patient registry is unclear, and validation of the registry has been considered necessary when ascertaining a stroke subtype as an endpoint [13].Taken together, the study collaborators found it scientifically sound to validate the diagnosis in a fairly large random sample. One hundred patients with a diagnosis of ischemic stroke were randomly selected for the validation procedure using patient record data. Among 100 patients, 89 had stroke and 87 had ischemic stroke. Two patients had intra-cerebral hemorrhage. It was unclear if one patient with fatal outcome had a stroke or not, and an autopsy was not undertaken. Of the 10 patients who did not have a stroke, four had a transitory ischemic attack (TIA) due to intra-cerebral thrombosis. Six patients did not have a cerebral ischemic event due to epilepsy (*n* = 1), primary progressive aphasia (*n* = 1), syncope (*n* = 1), disorientation (*n* = 1), headache (*n* = 1) and acute lower limb ischemia (*n* = 1). Among 87 with ischemic stroke, the distribution of causes was intra-cerebral thrombosis (*n* = 43; 49.4%), embolization secondary to atrial fibrillation (*n* = 31; 35.6%), embolization due to carotid artery stenosis (*n* = 7; 8.0%), carotid artery dissection (*n* = 2), embolization secondary to endocarditis (n = 1), unclear if symptomatic carotid artery stenosis or intra-cerebral thrombosis (*n* = 2) and unclear if cardiac arrhythmias or intra-cerebral thrombosis (*n* = 1). Among the 98 evaluable patients, 56 (57%) had an atherosclerotic cause of disease. The diagnosis of ischemic stroke was confirmed in 89% (87/98) of cases.

### 2.4. Non-Modifiable Risk Factors

Information on age and sex was extracted from the participants’ Swedish personal identification number. A heredity score for stroke was constructed based on the participants’ self-reported family history of stroke (mother, father, or sibling with stroke). Participants with no first-degree relatives (parent or sibling) with stroke were categorized having a low heredity for stroke while participants with at least two first-degree relatives were categorized as having high heredity for stroke. Educational level attained was defined as less than 9 years, elementary school (9–10 years), upper secondary school (11–13 years), university without a degree, and university degree.

### 2.5. Semi-Modifiable Risk Factors

Semi-modifiable risk factors were defined as comorbidities that can partly be modified by either pharmacological or lifestyle intervention. Hypertension was defined as systolic blood pressure ≥140 mmHg, diastolic blood pressure ≥90 mmHg or current use of antihypertensive medications. Prevalent diabetes mellitus was defined as having a measured fasting whole blood glucose ≥6.1 mmol/L, self-reported history of physician-diagnosed diabetes, use of diabetes medication, or being diagnosed and registered in any of the local or national diabetes registries. Serum concentrations of apolipoprotein-A-I (ApoA-I) and ApoB were measured by Quest Diagnostics (San Juan Capistraon, CA, USA) using an immunonephelometric assay run on the Siemens BNII (Siemens, Newark, DE, USA). The interassay coefficient of variability was <4.0% for both ApoA-I and ApoB. Dyslipidemia was subsequently defined as having an ApoB/ApoA-I ratio [14,15] of >0.9 for men and >0.8 for women or reporting current use of lipid-lowering drugs in the baseline questionnaire or in the 7-day menu book. Blood samples were collected from non-fasting participants at baseline. Prevalent atherosclerotic disease was defined as being diagnosed with coronary artery disease (ICD8-209, ICD9-414.9, ICD10-I25.9), peripheral arterial disease (ICD8-440.20/440.30, ICD9-443.9, ICD10-I70.2), or carotid artery disease (ICD8-432.90, ICD9-433.10, ICD10-I65.2) prior to baseline examinations by linkage to national registries.

### 2.6. Modifiable Risk Factors

Smoking was defined as never, former or current smoking. Weight (kg) and height (cm) were directly measured at baseline examinations and used to calculate body mass index (BMI; kg/m^2^). Participants were categorized as normal-weight (BMI < 25), overweight (BMI 25–29.9) or obese (BMI ≥ 30). Physical activity was categorized into low, moderate and high using sex-specific tertiles of a physical activity score based on reported time spent on leisure-time activities in the baseline questionnaire [16]. Subjects were categorized as zero consumers of alcohol if they reported no alcohol in their 7-day menu book and reporting no alcohol intake during the last year. The remaining participants were divided into sex-specific tertiles based on their reported intake of alcohol related to total energy intake (energy %). Dietary assessment was carried out using a modified diet history method. The participants filled in a 7-day menu book (including cooked meals, cold beverages, and medications/supplements) and a 168-item semi-quantitative food frequency questionnaire that included regularly consumed foods during the past year. Complementary information was gathered through 1 h interviews. We assessed diet quality using a previously developed diet quality index based on the Swedish nutrition recommendations [17] that was developed and validated for the MDCS cohort [18]. The index includes intake of six dietary components: saturated fatty acids ≤14 energy (E)%, polyunsaturated fatty acids 5–10 E%, fish and shellfish ≥300 g/week, sucrose ≤10 E%, dietary fiber ≥2.4 g/megajoule (MJ), fruit and vegetables ≥400 g/day. A reached recommendation results in one point per dietary component, with a maximum score of six points. Participants were categorized as having a low diet quality (0–1 points), a medium diet quality (2–4 points) and a high diet quality (5–6 points) [18].

### 2.7. Statistical Analysis

Baseline characteristics were expressed as mean and standard deviation (SD) for continuous variables, and as percentage of total count for categorical variables. Cox proportional hazards regression analysis was used to calculate hazard ratios (HR) with 95% confidence intervals (CI) with years of follow-up as the time scale. The basic model included adjustment for age and sex. A multivariable model including all examined risk factors in the same model was used to examine the independent effect of non-modifiable (age, sex, family history of stroke, and educational level), semi-modifiable (hypertension, dyslipidemia, diabetes mellitus, and atherosclerotic disease), and modifiable (smoking, diet quality, physical activity, BMI, and alcohol intake) risk factors. The proportional hazards assumption for presented models were tested using the Schoenfeld residuals test. No deviations were noted. To examine whether the effect of modifiable lifestyle risk factors on atherothrombotic stroke differ by the semi-modifiable comorbidities at baseline we tested for multiplicative interaction by including the cross-product (lifestyle factor × comorbidity) in the model. The 10-year atherothrombotic stroke rates by strata of lifestyle factors were calculated using a Cox regression model and standardized to the mean of all predictor variables, including other lifestyle risk factors as well as the non-modifiable and semi-modifiable risk factors. To examine the predictive utility of non-modifiable, semi-modifiable, and modifiable risk factors for atherothrombotic stroke we calculated Harrell’s concordance statistic (C-index) and specifically examined the added benefit of modifiable lifestyle risk factors to the model including non-modifiable and semi-modifiable risk factors using the likelihood ratio test. For statistical analyses IBM SPSS Statistics (Version 26, Chicago, IL, USA), Stata/SE (Version 14.2, College Station, TX, USA) and R (Version 3.5.1, The R Project for Statistical Computing, Vienna, Austria) were used. All tests were two-sided and statistical significance level was set at *p* < 0.05 for the primary analyses. For tests of interaction between lifestyle factors and comorbidities, we corrected for multiple testing using the Bonferroni method and statistical significance level was set at *p* < 0.0025.

## 3. Results

### 3.1. Description of the Study Population

After a median of 21.2 years (interquartile range (IQR) 7.0) of follow-up, 2339 out of 26,547 (8.8%) patients were diagnosed with atherothrombotic stroke. The total incidence rate per 1000 person-years of follow-up was 4.59 (95% CI: 4.41–4.78) and the 10-year incidence rate per 1000 person-years of follow-up was 3.52 (95% CI: 3.29–3.76). Baseline characteristics for those with and without incident atherothrombotic stroke are presented in Table 1. Individuals with future atherothrombotic stroke were at baseline older, more likely to be male, had a higher stroke heredity, lower educational level, higher BMI, consumed less alcohol, were more likely to be current smokers, were less physically active and had a lower diet quality. Prevalent hypertension, dyslipidemia, diabetes mellitus and atherosclerotic disease was more common in the group with incident atherothrombotic stroke.

### 3.2. Non-Modifiable and Semi-Modifiable Risk Factors for Atherothrombotic Stroke

A summary of the main findings of independent risk factors for atherothrombotic stroke can be found in Table 2. In the multivariable-adjusted model, all established non-modifiable risk factors were associated with incident atherothrombotic stroke (Table 1). We observed higher risk associated with age (HR per year = 1.08, 95% CI 1.07–1.08), male gender (HR 1.41, 95% CI 1.29–1.54), having at least two first-degree relatives with stroke (HR 1.25, 95% CI 1.04–1.51) and a lower risk among those with a university degree (HR 0.85, 95% CI 0.74–0.98) (Table 1). At-risk comorbidities at baseline were also associated with higher risk of atherothrombotic stroke including hypertension (HR 1.49; 95% CI 1.35–1.64), dyslipidemia (HR 1.22; 95% CI 1.12–1.34), diabetes mellitus (HR 2.17; 95% CI 1.87–2.50) and atherosclerotic disease (HR 1.49; 95% CI 1.21–1.83) (Table 1). The C-statistic for a model including the non-modifiable risk factors was 0.6888 and additional inclusion of semi-modifiable risk factors significantly increased the C-statistic to 0.7065 (*p* < 0.00001).

### 3.3. Modifiable Lifestyle Risk Factors for Atherothrombotic Stroke

Among the modifiable lifestyle risk factors, non-smoking (never or former) was compared to current smoking associated with a reduced risk of incident atherothrombotic stroke (HR never smokers compared to current smokers = 0.62, 95% CI 0.56–0.58). Having a normal weight (BMI < 25 was compared to being obese (BMI > 30) associated with a lower risk of atherothrombotic stroke (HR = 0.83, 95% CI 0.73–0.94) after adjustment for age and sex. In the multivariable model there was a null association between BMI and atherothrombotic stroke. A high level of physical activity (HR 0.89, 95% CI 0.80–0.98) and a high diet quality (HR 0.83; 95% CI 0.72–0.97) were associated with a decreased risk of atherothrombotic stroke (Table 1). In an age- and sex-adjusted model, zero alcohol consumers were at higher risk of atherothrombotic stroke compared to low alcohol consumers, and moderate alcohol consumption was associated with a lower risk. These associations were however attenuated and not statistically significant in the multivariable analysis (Table 1). Including all five modifiable lifestyle factors in addition to non-modifiable and semi-modifiable risk factors in a Cox model for atherothrombotic stroke significantly increased the C-statistic to 0.7181 (*p* < 0.00001). When including only the statistically significant lifestyle factors (i.e., smoking, diet and physical activity) to the model including non-modifiable and semi-modifiable risk factors the C-statistic was 0.7172 (*p* < 0.00001).

We further examined whether observed associations between modifiable lifestyle risk factors and atherothrombotic stroke differed among subjects with and without baseline comorbidities associated with a higher risk of stroke (Appendix A). There was no statistically significant heterogeneity by comorbidities for the observed associations (all P-interaction > 0.10).

We estimated the standardized 10-year risk of atherothrombotic stroke among subjects stratified by lifestyle risk factors. The relative risk reductions observed with favorable lifestyle factors corresponded to a decrease in absolute 10-year risk from 3.8% (95% CI: 3.4–4.1) among current smokers to 2.3% (95% CI: 2.1–2.5) among never smokers; from 3.0% (95% CI: 2.7–3.4) among those with low diet quality to 2.4% (95% CI: 2.1–2.8) among those with high diet quality; and from 2.9% (95% CI: 2.6–3.2) among those with low physical activity to 2.6% (95% CI: 2.3–2.8) among those with high physical activity.

## 4. Discussion

In this population-based prospective cohort study with a median follow-up of 21.2 years we found that established risk factors for ischemic stroke [19] are also independently associated with atherothrombotic stroke. Among the modifiable lifestyle factors examined, non-smoking, high physical activity and a high diet quality were independently associated with lower risk of atherothrombotic stroke. There was no significant interaction between modifiable risk factors and comorbidities, indicating a similar effect of a favorable lifestyle on the risk of atherothrombotic stroke regardless of underlying comorbidities.

According to the World Health Organization (WHO), up to 80% of non-communicable diseases including cardiovascular disease, type 2 diabetes and cancer are preventable by eliminating shared modifiable risk factors; smoking, unhealthy diet, physical inactivity and harmful use of alcohol [20]. Both hypertension and diabetes mellitus are considered important risk factors for all stroke subtypes [21], a result that previously has been shown within the MDCS [22,23], and was reproduced in the present study on incident atherothrombotic stroke. Blood pressure control has been proven to be an effective preventive measure of stroke in individuals both with and without diabetes [24]. In addition, lowering blood glucose and lipids through lifestyle changes or medications are other examples of effective stroke prevention. A randomized controlled trial (RCT) on intensive lifestyle intervention aiming for weight loss was compared with usual care among 5145 obese diabetics and found no significant difference regarding cardiovascular morbidity and mortality after 9.6 years of follow-up [25]. This points to the fact that lifestyle interventions entirely aiming for weight loss do not lead to an equivalent decrease in CVD incidence. This is coherent with the results from the present study, where no significant association between high BMI and increased risk of atherothrombotic stroke was found. However, abdominal obesity was shown to be one of the five risk factors that together accounted for 80% of risk for stroke in a large case-control study including 6000 patients [21], and is probably a more relevant anthropometric measurement in this respect.

In the PREDIMED study a Mediterranean diet with nuts or extra virgin olive oil was associated with a reduced risk of stroke compared to a low-fat control diet [26]. A large systematic review on 142 prospective cohort studies showed that a high fruit and vegetable intake, up to 800 g/day, reduces the risk for stroke and cardiovascular disease [27]. Previous studies within the MDCS have also found that a high diet quality based on adherence to the Swedish nutrition recommendations associate with lower risk of incident cardiovascular disease [28] all-cause and cardiovascular mortality [29]. The findings for an association between overall diet quality and atherothrombotic stroke require further study to examine if there are specific diet components that may drive this association. The protective effect of physical activity on stroke could be explained by its ability to reduce blood pressure, blood lipids, excess body weight and preventing diabetes mellitus [30]. Physical activity in midlife was recently shown to be associated with reduced incidence of vascular dementia in two separate cohorts, among long-distance skiers and in the MDCS, respectively, an association that could not be shown for subsequent development of Alzheimer’s disease [31]. These results are in line with the present study, which highlight the importance of a healthy lifestyle for reducing the risk of atherothrombotic ischemic stroke, regardless of prevalent comorbidities.

The association between alcohol consumption and incidence of cardiovascular diseases is often described as a J-shaped dose-response relationship with lower risk for light-to-moderate alcohol drinkers [32]. The absence of increased risk of high alcohol consumption and incident atherothrombotic stroke might be due to the fact that participants were comparably healthier than non-participants [33] and to reporting bias. The observation that zero alcohol consumers had an increased risk of incident atherothrombotic stroke in the age- and sex adjusted model and a trend for increased risk in the fully adjusted model is probably due to residual confounding. Abstainers may include subjects who were formerly heavy drinkers; abstainer bias, and some participants with serious illness may have quit drinking alcohol—sick quitter bias [34].

A high heredity score based on the number of first-degree relatives with stroke was compared to a low heredity score associated with a 25% increased risk of incident atherothrombotic stroke in the present study. Interestingly, an extensive polygenic risk score based on evaluation of single nucleotide polymorphisms has been found to add important predictive information in relation to ischemic stroke in addition to established risk factors, including family history of stroke [35]. Furthermore, a polygenic risk score of 90 single nucleotide polymorphisms and a lifestyle score were recently found to independently associate with incident stroke in a stroke population where 74% had ischemic stroke [11]. This finding implies a benefit of populations adhering to a healthy lifestyle, regardless of their underlying genetic risk.

There are several weaknesses that warrant discussion. Firstly, there is a risk of misreporting bias when it comes to self-reported dietary and lifestyle habits. The modified diet history method used in the MDCS has however been shown to have a high relative validity [36,37]. We further assessed diet quality in this study based on a previously developed diet quality index shown to have high content validity [18]. Furthermore, for socio-demographic and lifestyle variables collected through the baseline questionnaire, the agreement between the baseline questionnaire and the same questionnaire when repeated after 3 weeks was high for most variables (kappa > 0.75) [38]. The participation rate in the eligible population was 40%, possibly leading to selection bias and reducing the generalizability of the results. A health survey was mailed to the same population, which resulted in a 74.6% participant rate. The survey showed a comparable socio-demographic structure between participants and non-participants, but the number of participants reporting good health was higher in the MDCS cohort [33]. The current study is observational in its nature and thus reported statistical associations may not represent causal associations. Furthermore, the aim of this study was only to identify independent predictors of atherothrombotic stroke and not to build models more suitable to address the issue of causality. However, in support of our findings, a recent Mendelian randomization study found that genetically predicated years of education were inversely associated with several ischemic stroke subtypes [39]. Further genetically predicted smoking and body mass index increased risk of ischemic and large artery stroke. In this study, we observed a positive association between obesity and atherothrombotic stroke after adjustment for age and sex, but a null association after adjustment for other established risk factors. However, the main predictors of obesity (diet and physical activity) were found to be independently associated with atherothrombotic stroke, suggesting a potential beneficial effect not mediated by their effect on body mass index. Thus, overall, our findings are in line with the Mendelian randomization results suggesting that modifications of obesity through diet and physical activity, as well as smoking cessation are important targets for the primary prevention of atherothrombotic stroke. It should be noted that the current study population was followed-up for a considerable time period, however, both semi-modifiable and modifiable risk factors were assessed at baseline only. Repeated measurements would lower the risk of potential residual confounding by these factors [40]. A change in classification over time is, however, likely to result in non-differential misclassification which would tend to bias the observed associations towards the null. Finally, it should be noted that this study was underpowered to fully address the issue of heterogeneity by some of the baseline comorbidities for the association between lifestyle factors and atherothrombotic stroke. In particular, there was a limited number of participants with prevalent diabetes mellitus (4.2% of the study population) and atherosclerotic disease (2.3% of the study population) resulting in low statistical power.

The main strengths of the present study include the large study population and a follow-up time of more than 20 years. The fact that registries were used to identify endpoints ensured nearly no loss to follow-up (<0.5%). The ascertainment of AF in the MCDS cohort has previously been shown to have high validity after scrutinizing electrocardiographs [41]. The present study had access to an extensive data set, enabling multiple confounders to be taken into consideration. Furthermore, the validity of the ischemic stroke diagnosis was high, with confirmation of the diagnosis in 89% of the validation sample. Ischemic stroke preceded by AF is most likely caused by a cardiac embolism and not by thrombosis secondary to severe atherosclerosis. To ensure adherence to the endpoint atherothrombotic stroke and avoid inclusion of cardioembolic stroke, patients with AF prior to or simultaneous to their ischemic stroke were adjusted for during follow-up.

## 5. Conclusions

In conclusion, the present study found that the effect of modifiable lifestyle risk factors on the risk of atherothrombotic ischemic stroke is similar regardless of underlying comorbidities. A healthy lifestyle including non-smoking, a high diet quality and a high level of physical activity was shown to significantly reduce the risk of atherothrombotic stroke. These findings confirm the potential utility of current recommendations for cardiovascular disease prevention in the prevention of atherothrombotic ischemic stroke.

## Figures and Tables

**Figure 1 nutrients-13-01952-f001:**
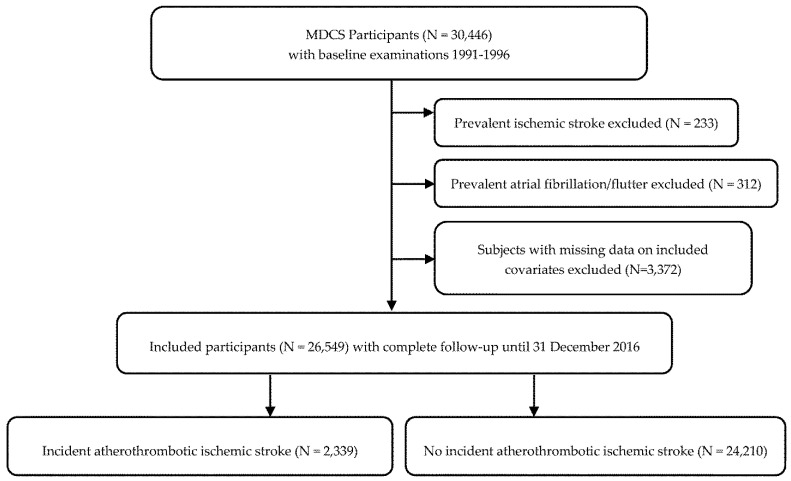
Descriptive flow diagram of study participants and exclusions. Some individuals had multiple exclusion criteria.

**Table 1 nutrients-13-01952-t001:** Non-modifiable, semi-modifiable, and modifiable risk factors ^1^ in relation to incident atherothrombotic ischemic stroke among subjects of the Malmö Diet and Cancer Study (N = 26,549). Hazard ratios (HR) and 95% confidence intervals from Cox proportional hazards regression models.

Risk Factors	All Subjects N = 26,549	No Athero-Thrombotic Stroke N = 24,210	Athero-Thrombotic Stroke N = 2339	Age- and Sex Adjusted HR (95% CI)	*p*-Value	Multivariable Adjusted HR (95% CI) ^2^	*p*-Value
Non-modifiable							
Age, years	58.01 (7.62)	57.7 (7.6)	61.4 (7.1)	1.09 (1.08–1.09)	2.1 × 10^−174^	1.08 (1.07–1.08)	1.0 × 10^−123^
Male sex, %	38.8	38.0	47.8	1.49 (1.38–1.62)	4.9 × 10^−22^	1.41 (1.29–1.54)	6.5 × 10^−15^
Stroke heredity score, %							
0 (lowest)	73.1	73.5	69.0	1.00 (ref)		1.00 (ref)	
1	23.3	23.1	25.9	1.12 (1.02–1.23)	0.17	1.12 (1.02–1.23)	0.022
2 (highest)	3.6	3.5	5.2	1.30 (1.08–1.57)	5.5 × 10^−3^	1.25 (1.04–1.51)	0.017
Educational level, %							
Less than 9 years	41.6	40.7	51.1	1.00 (ref)		1.00 (ref)	
Elementary school (9–10 years)	26.2	26.6	23.0	0.81 (0.73–0.90)	5.0 × 10^−5^	0.86 (0.77–0.95)	3.4 × 10^−3^
Elementary + upper secondary school (9–13 years)	9.0	9.1	7.8	0.80 (0.68–0.94)	5.4 × 10^−3^	0.87 (0.74–1.02)	0.084
University studies, no degree	8.8	8.9	7.8	0.83 (0.71–0.97)	0.020	0.92 (0.78–1.07)	0.28
University studies, with degree	14.4	14.8	10.2	0.74 (0.64–0.85)	2.6 × 10^−5^	0.85 (0.74–0.98)	0.026
Semi-modifiable							
Hypertension, %	61.1	59.7	75.4	1.55 (1.41–1.71)	1.2 × 10^−18^	1.49 (1.35–1.64)	3.3 × 10^−15^
Dyslipidemia, %	23.3	22.6	31.0	1.40 (1.28–1.52)	2.6 × 10^−14^	1.22 (1.12–1.34)	8.6 × 10^−6^
Diabetes mellitus, %	4.2	3.7	8.7	2.28 (1.97–2.64)	3.8 × 10^−29^	2.17 (1.87–2.50)	7.8 × 10^−25^
Atherosclerotic disease, %	2.3	2.1	4.3	1.73 (1.41–2.12)	1.2 × 10^−7^	1.49 (1.21–1.83)	1.3 × 10^−4^
Modifiable							
Smoking, %							
Current	28.1	27.8	31.9	1.00 (ref)		1.00 (ref)	
Former	33.8	33.9	32.0	0.62 (0.56–0.69)	9.6 × 10^−20^	0.62 (0.56–0.69)	8.7 × 10^−19^
Never	38.1	38.3	36.0	0.60 (0.55–0.67)	8.3 × 10^−23^	0.62 (0.56–0.68)	3.0 × 10^−20^
Body mass index, %							
30>	13.0	12.9	14.7	1.00 (ref)		1.00 (ref)	
25–29.9	39.4	38.9	44.0	0.92 (0.81–1.04)	0.182	1.03 (0.91–1.17)	0.62
<25	47.6	48.2	41.3	0.83 (0.73–0.94)	2.6 × 10^−3^	0.98 (0.86–1.12)	0.80
Diet quality, %							
Low	15.2	15.1	16.4	1.00 (ref)		1.00 (ref)	
Medium	71.3	71.4	70.6	0.86 (0.77–0.96)	7.8 × 10^−3^	0.89 (0.80–1.00)	0.051
High	13.4	13.5	13.0	0.80 (0.69–0.93)	3.6 × 10^−3^	0.83 (0.72–0.97)	0.022
Physical activity, %							
Low	32.9	32.7	35.0	1.00 (ref)		1.00 (ref)	
Moderate	33.6	33.8	32.1	0.85 (0.77–0.94)	1.3 × 10^−3^	0.91 (0.82–1.00)	0.055
High	33.5	33.5	32.9	0.82 (0.74–0.90)	5.9 × 10^−5^	0.89 (0.80–0.98)	0.018
Alcohol consumption, %							
Zero	6.0	5.8	7.8	1.24 (1.05–1.46)	9.5 × 10^−3^	1.16 (0.99–1.37)	0.067
Low	30.9	30.6	33.7	1.00 (ref)		1.00 (ref)	
Moderate	31.5	31.7	29.5	0.85 (0.77–0.95)	2.4 × 10^−3^	0.91 (0.82–1.00)	0.058
High	31.6	31.8	28.9	0.95 (0.86–1.06)	0.373	0.99 (0.89–1.11)	0.92

^1^ Continuous variables presented as mean (standard deviation, SD) and categorical variables as percentage of total count. ^2^ Multivariable model including all risk factors in the table.

**Table 2 nutrients-13-01952-t002:** Summary of main findings. The table shows the independent non-modifiable, semi-modifiable, and modifiable risk factors for atherothrombotic ischemic stroke identified in this population-based study.

Non-Modifiable	Semi-Modifiable	Modifiable
Age	Hypertension	Current smoking
Sex	Dyslipidemia	Low diet quality
Family history of stroke	Diabetes mellitus	Low physical activity level
Low educational level	Atherosclerotic disease	

## Data Availability

Restrictions apply to the availability of these data. Data was obtained from The Malmö Cohorts and are available at https://www.malmo-kohorter.lu.se/malmo-cohorts with the permission of MDC Steering Committee.

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
