# Peer review of "Modifiable and Non-Modifiable Risk Factors for Atherothrombotic Ischemic Stroke among Subjects in the Malmö Diet and Cancer Study"

_nutrients, 2021, doi:10.3390/nu13061952_

Round 1
Reviewer 1 Report
Authors have made an interesting study about an important subject. The study is well designed and put in the practise. However, there are few little things that need to be considered.
1. Please write out what abbreviation MDCS means when it is presented first time in the text (Materials and Methods, line 76).
2. Please add some discussion why it might be that alcohol consumption was not associated to atherothrombotic ischemic stroke in this study. Could reporting bias explain this? Is it possible that some of zero alcohol consumers were former alcoholics? Might this explain your observation that in an age- and sex-adjusted model, zero alcohol consumers were at higher 265 risk of atherothrombotic stroke compared to low alcohol consumers.
3. Please add some discussion that could diet score (and thus also Swedish nutrition recommendations) be somehow better? Now recommendations allow red and processed meat 500g a week. They also encourage to reduce amount of added sugar but they don´t say "don´t eat sugar". Might further reduction (or total avoidance) of red and processed meat and/or sugar lead even better results?
4. Studies have also shown that eating fruits and vegetables 800g/day is even better than 400g/day. Please add some discussion also about this. https://www.ncbi.nlm.nih.gov/pmc/articles/PMC5837313/
Author Response
Reviewer 1
Authors have made an interesting study about an important subject. The study is well designed and put in the practise. However, there are few little things that need to be considered.
- Please write out what abbreviation MDCS means when it is presented first time in the text (Materials and Methods, line 76).
Thanks. Done.
- Please add some discussion why it might be that alcohol consumption was not associated to atherothrombotic ischemic stroke in this study. Could reporting bias explain this? Is it possible that some of zero alcohol consumers were former alcoholics? Might this explain your observation that in an age- and sex-adjusted model, zero alcohol consumers were at higher risk of atherothrombotic stroke compared to low alcohol consumers.
Thanks, we discuss the relationship between alcohol and incidence of cardiovascular disease in a new paragraph in the Discussion, page 8, line 372 – 382:
The association between alcohol consumption and incidence of cardiovascular diseases is often described as a J-shaped dose-response relationship with lower risk for light-to-moderate alcohol drinkers (ref Yoon). The absence of increased risk of high alcohol consumption and incident atherothrombotic stroke might be due to that participants were comparably healthier than non-participants (ref Manjer) and to reporting bias. The observation that zero alcohol consumers had an increased risk of incident atherothrombotic stroke in the age- and sex adjusted model and a trend for increased risk in the fully adjusted model is probably due to residual confounding. Abstainers may include subjects that were formerly heavy drinkers, abstainer bias, and some participants with serious illness may have quit drinking alcohol, sick quitter bias (ref Xi).
Refs:
Yoon S-J, Jung J-G, Lee S, Kim J-S, Ahn S-K, Shin E-S, Jang J-E, Lim S-H. The protective effect of alcohol consumption on the incidence of cardiovascular diseases: is it real? A systematic review and meta-analysis of studies conducted in community settings. BMC Public Health 2020 20:90.
Xi Bo, Veeranki SP, Zhao M, Ma C, Yan Y, Mi J. Relationship of alcohol consumption to all-cause, cardiovascular, and cancer-related mortality in US. Adults. J Am Coll Cardiol 2017; 70: 913 – 22.
- Please add some discussion that could diet score (and thus also Swedish nutrition recommendations) be somehow better? Now recommendations allow red and processed meat 500g a week. They also encourage to reduce amount of added sugar but they don´t say "don´t eat sugar". Might further reduction (or total avoidance) of red and processed meat and/or sugar lead even better results?
The diet quality index was developed and validated in the Malmö Diet and Cancer cohort prior to the publication on the 2012 Nordic Nutrition Recommendations. Thus, red and processed meat intake was not included as a dietary component in the dietary index. The dietary fat components (in particular SFA) was useful in discriminating intake of meats and high-fat dairy. The sucrose component was identified as an important marker of consumption of discretionary foods, such as sugary foods and drinks. The question posed by the reviewer – i.e. “if the recommendations would be more strict, is there an additional beneficial effect on risk of atherothrombotic stroke?”. Unfortunately, this question cannot be answered in the current study. The reason for this is that there are no or very few subjects reporting zero or very low intake of meats or sugary foods/drinks. In this study, we limited the investigation into diet to overall diet quality which has previously in this cohort been linked to higher incidence and mortality from cardiovascular disease (overall) to examine if diet may associate also with specifically atherothrombotic ischemic stroke. A detailed discussion into the current nutrition recommendations and specific aspects of diet (such as red meat or sugar drinks) we find to be out of scope for the current study. We have however added the following in the discussion, p 8, lines 358-360: “The findings for an association between overall diet quality and atherothrombotic stroke require further study to examine if there are specific diet components that may drive this association.”
- Studies have also shown that eating fruits and vegetables 800g/day is even better than 400g/day. Please add some discussion also about this. https://www.ncbi.nlm.nih.gov/pmc/articles/PMC5837313/
Thanks, this reference was already referred to in the discussion, page 8, lines 360 – 362. We simply added: “A large systematic review on 142 prospective cohort studies showed that a high fruit and vegetable intake, up to 800 g/day, reduces the risk for stroke and cardiovascular disease [27].”
Using predefined cut-offs based on current nutrition recommendations are easier to interpret. We do however acknowledge that there is a certain loss of information by dichotomizing the included diet components, such as fruits and vegetables, since the added benefit of very high intakes may be diluted. A detailed discussion regarding the specific components used to classify individuals into groups of low versus high overall diet quality is however out of scope for this study. The added benefit of very high intakes of fruits and vegetables is further difficult to examine in this study population since only a very small proportion (2.7%) of the population reach such high intake levels.
Reviewer 2 Report
It is recommended to divide table 1 into 3 tables:
Non-modifiable, semi-modifiable, and modifiable risk factors
It is recommended to add this quote: https://doi.org/10.3390/jcm9072233
in this paragraph:
Several observational studies have reported that an overall healthy lifestyle reduces the 55 risk of ischemic stroke using lifestyle scores including non-smoking, absence of 56 overweight, moderate physical activity and alcohol consumption, and a high diet quality 57 [8,9, 10].
Author Response
Reviewer 2
It is recommended to divide table 1 into 3 tables:
Non-modifiable, semi-modifiable, and modifiable risk factors
Thanks for the suggestion. We hope the publisher can help us to adjust this table to a more attractive and easy survey table.
It is recommended to add this quote: https://doi.org/10.3390/jcm9072233
in this paragraph:
Several observational studies have reported that an overall healthy lifestyle reduces the risk of ischemic stroke using lifestyle scores including non-smoking, absence of overweight, moderate physical activity and alcohol consumption, and a high diet quality [8,9,10].
Thanks for the suggestion of reference by Soto-Camara, but there was no data on diet in the paper.
Reviewer 3 Report
The paper is very interesting and it explores a very important topic about healthcare system and patients' quality of life. Thank you to cite the limits of the study, moreovere it involves a very large number of subjects.
Some concerns:
It would be important to analyze the role of some genetic factors, as the genetic variant of RANKL, as an independent risk factor for ischemic stroke [RANK/RANKL/OPG pathway: genetic association with history of ischemic stroke in Italian population European review for medical and pharmacological sciences 20, Issue 21, 1 November 2016, Pages 4574-4580] and about the high risk of recurrent stroke among patient with symptomatic intracranial atherosclerotic disease [Symptomatic intracranial atherosclerotic disease: an ultrasound 2-year follow-up pilot study. Neurological Sciences Vol.39, Issue 11, 1 November 2018, Pages 1955-1959].
About Nutrients authors should cite the role of some of those involved in the process expecially in elderly (that represents the majority of stroke patients) [Selenium Concentrations and Mortality Among Community-Dwelling Older Adults: Results from ilSIRENTE Study. Journal of Nutrition, Health and Aging, 2018, 22(5), pp. 608–612 - Myeloperoxidase levels and mortality in frail community-living elderly individuals. Journals of Gerontology - Series A Biological Sciences and Medical Sciences, 2010, 65 A(4), pp. 369–376].
Finally, authors should consider the important role of prevention and of Rehabilitation, in stroke, during a particular period of pandemic [Global approaches for global challenges: The possible support of rehabilitation in the management of COVID-19. Journal of Medical Virology, 2020, 92(10), pp. 1739–1740].
Author Response
Reviewer 3
The paper is very interesting and it explores a very important topic about healthcare system and patients' quality of life. Thank you to cite the limits of the study, moreover it involves a very large number of subjects.
Some concerns:
It would be important to analyze the role of some genetic factors, as the genetic variant of RANKL, as an independent risk factor for ischemic stroke [RANK/RANKL/OPG pathway: genetic association with history of ischemic stroke in Italian population European review for medical and pharmacological sciences 20, Issue 21, 1 November 2016, Pages 4574-4580] and about the high risk of recurrent stroke among patient with symptomatic intracranial atherosclerotic disease [Symptomatic intracranial atherosclerotic disease: an ultrasound 2-year follow-up pilot study. Neurological Sciences Vol.39, Issue 11, 1 November 2018, Pages 1955-1959].
About Nutrients authors should cite the role of some of those involved in the process expecially in elderly (that represents the majority of stroke patients) [Selenium Concentrations and Mortality Among Community-Dwelling Older Adults: Results from ilSIRENTE Study. Journal of Nutrition, Health and Aging, 2018, 22(5), pp. 608–612 - Myeloperoxidase levels and mortality in frail community-living elderly individuals. Journals of Gerontology - Series A Biological Sciences and Medical Sciences, 2010, 65 A(4), pp. 369–376].
Finally, authors should consider the important role of prevention and of Rehabilitation, in stroke, during a particular period of pandemic [Global approaches for global challenges: The possible support of rehabilitation in the management of COVID-19. Journal of Medical Virology, 2020, 92(10), pp. 1739–1740].
Thanks for all these references, which we have read with interest. However, we find it difficult to make them fit into the present study.
Reviewer 4 Report
The study is a population-based cohort study which aimed to examine the associations between modifiable and non-modifiable risk factors and atherothrombotic stroke, and to examine if the potential benefit of modifiable lifestyle factors differs among subjects with and without predisposing comorbidities. There are several issues need to be concerned.
- Suggest that the authors should provide the numbers in Table 2 not only summary, which is more clearly to understand. The results from line 246 ~ 254 and line 285 ~ 291 should be shown using tables or figures.
- In addition, the results showed C-statistics but the purpose of the study did not mention about the predictability of these risk factors.
- Materials and Methods section: line 76 ~ 97 might be merged together which is much better to understand. The description suggests to be written according to the flow chart in Figure 1.
- The results of this study come from registry data, which I understand should be more accurate than the collected database. It’s quite confused why need to validate the diagnosis of ischemic stroke. Or the author need to explain the accuracy of the registry system.
- For the definition of dyslipidemia, it is unusual to use ApoAI and Apo B. Why not use the common criteria of dyslipidemia?
- Line 198, suggest to confirm reference 8 is correct or not.
- The study subjects whose age is 57.7 and the percentage of DM is 4.2%, which is quite low. The results might be biased due to healthy subject effect.
- This study followed for 21.2 years. The semi-modifiable and modifiable risk factors might be changed overtime which might be biased the results. Although the authors provided the agreement between baseline questionnaire and the same questionnaire when repeated after 3 weeks, the duration of the review is far less than 20 years.
Author Response
Reviewer 4
The study is a population-based cohort study which aimed to examine the associations between modifiable and non-modifiable risk factors and atherothrombotic stroke, and to examine if the potential benefit of modifiable lifestyle factors differs among subjects with and without predisposing comorbidities. There are several issues need to be concerned.
- Suggest that the authors should provide the numbers in Table 2 not only summary, which is more clearly to understand. The table is meant as a summary table. For numbers, please see Table 1.
- The results from line 246 ~ 254 and line 285 ~ 291 should be shown using tables or figures. These results (lines 246 – 254) are highlighted in the text and are shown in Table 1. We believe the lines 285 – 291 fits as a stand alone text paragraph without table.
- In addition, the results showed C-statistics but the purpose of the study did not mention about the predictability of these risk factors. C-statistic was applied to evaluate the secondary aim in how much the life-style factors contributed to improved prediction apart from the established risk factors. We added a secondary aim in the last part of the introduction to clarify this on page 2, line 77-79: “A secondary aim was to investigate to what extent modifiable risk factors contribute to improved prediction in addition to established risk factors.”
- Materials and Methods section: line 76 ~ 97 might be merged together which is much better to understand. The description suggests to be written according to the flow chart in Figure 1. The flow chart relates to the first paragraph, “Study population and data collection”, whereas the second paragraph, endpoint ascertainment, includes adjustment for atrial fibrillation or flutter during follow-up time. We would like to keep this structure.
- The results of this study come from registry data, which I understand should be more accurate than the collected database. It’s quite confused why need to validate the diagnosis. of ischemic stroke. Or the author need to explain the accuracy of the registry system. The prospective database with baseline data is very accurate and inserted into a predefined database. The endpoint data during follow-up is retrieved from the in-hospital patient registry system.
We add in page 4, line 138-142: "Validity of diagnosis of ischemic stroke in the national patient registry is unclear, and validation of the registry has been considered necessary when ascertaining a stroke subtype as an endpoint (ref Zia). Taken together, the study collaborators found it scientifically sound to validate the diagnosis in a fairly large random sample."
Ref:
Zia E, Hedblad B, Pessah-Rasmussen H, Berglund G, Janzon L, Engström G. Blood pressurein relation to the incidence of cerebral infarction and intracerebral hemorrhage. Stroke 2007; 38: 2681 – 2685.
- For the definition of dyslipidemia, it is unusual to use ApoAI and Apo B. Why not use the common criteria of dyslipidemia? As stated in Methods – Semi-modifiable risk factors: “Dyslipidemia was subsequently defined as having an ApoB/ApoA-I ratio (Refs) of >0.9 for men and >0.8 for women or reporting current use of lipid-lowering drugs in the baseline questionnaire or in the 7-day menu book. Blood samples were collected from non-fasting participants at baseline.”
The main reason for using this measure of dyslipidemia was that ApoB and ApoA-I were measured in full cohort, while other markers of dyslipidemia such as LDL, HDL and triglycerides are only available in a smaller subset of the population (approx. 5000 subjects) which would result in a strong reduction in statistical power. While the ApoB/ApoA-I ratio is not the most common definition of dyslipidemia it may in fact be a better marker of cardiovascular disease than LDL, HDL and the LDL/HDL ratio. ApoB concentrations are a measure of the total numbers of all atherogenic particles, including VLDL, IDL and LDL, while ApoA-I is a measure of HDL particles.
We added two references in Methods to support the use of ApoB/ApoA-I ratio.
Refs:
Walldius G, Jungner I, Aastveit AH, Holme I, Furberg CD, Sniderman AD. The apoB/apoA-I ratio is better than the cholesterol ratios to estimate the balance between plasma proatherogenic and antiatherogenic lipoproteins and to predict coronary risk. Clin Chem lab Med 2004; 42: 1355 – 1363.
Walldius G, Jungner I. The apob/apoa-I ratio: A strong new risk factor for cardiovascular disease and a target for lipid-lowering therapy – A review of the evidence. J Intern med 2006; 259: 493 – 519.
- Line 198, suggest to confirm reference 8 is correct or not.
- Thanks. It was not correct. It should have been reference 15 and we revised accordingly.
- The study subjects whose age is 57.7 and the percentage of DM is 4.2%, which is quite low. The results might be biased due to healthy subject effect.
- Agree. We discuss this possible healthy subject effect in Discussion, lines 364 – 369. “The participation rate in the eligible population was 40%, possibly leading to selection bias and reducing the generalizability of the results. A health survey was mailed to the same population, which rendered in a 74.6% participant rate. The survey showed a comparable socio-demographic structure between participants and non-participants, but the number of participants reporting good health was higher in the MDCS cohort [33].”
- This study followed for 21.2 years. The semi-modifiable and modifiable risk factors might be changed overtime which might be biased the results. Although the authors provided the agreement between baseline questionnaire and the same questionnaire when repeated after 3 weeks, the duration of the review is far less than 20 years.
- Agree. We added in Discussion, page 9, line 413-419:
It should be noted that the current study population was followed-up for a considerable time period, however, both semi-modifiable and modifiable risk factors were assessed at baseline only. Repeated measurements would lower the risk of potential residual confounding by these factors (Ref). A change in classification over time is, however, likely to result in non-differential misclassification which would tend to bias the observed associations towards the null.
Ref:
Alhadad A, Wictorsson C, Alhadad H, Lindblad B, Gottsäter A. Medical risk factor treatment in peripheral arterial disease – need for further improvement. Int Angiol 2013; 32: 332 – 338.
Round 2
Reviewer 3 Report
Sorry, I think genetic factors shoud be cited
Author Response
Thanks. We included the following paragraph in the discussion:
"A high heredity score based on the number of first-degree relatives with stroke was compared to a low heredity score associated with a 25% increased risk of incident atherothrombotic stroke in the present study. Interestingly, an extensive polygenic risk score based on evaluation of single nucleotide polymorphisms has been found to add important predictive information in relation to ischemic stroke in addition to established risk factors, including family history of stroke (ref A). Further, a polygenic risk score of 90 single nucleotide polymorphisms and a lifestyle score were recently found to independently associate with incident stroke in a stroke population where 74% had ischemic stroke [11]. This finding implies a benefit of populations adhering to a healthy lifestyle, regardless of their underlying genetic risk."
ref A
Abraham G, Malik R, Yonova-Doing E, Salim A, Wang T, Danesh J, Butterworth AS, Howson JMM, Inouye M, Dichgans M. Genomic risk score offers predictive performance comparable to clinical risk factors for ischaemic stroke. Nat Commun. 2019 Dec 20;10(1):5819. doi: 10.1038/s41467-019-13848-1. Erratum in: Nat Commun. 2020 Feb 20;11(1):1036. PMID: 31862893; PMCID: PMC6925280.
Reviewer 4 Report
The authors revised the manuscript point-by point accordingly. I think the study now fulfill the publication standard.
Author Response
Thank you.